# Group Target Tracking for Highly Maneuverable Unmanned Aerial Vehicles Swarms: A Perspective

**DOI:** 10.3390/s23094465

**Published:** 2023-05-04

**Authors:** Yudi Chen, Yiwen Jiao, Min Wu, Hongbin Ma, Zhiwei Lu

**Affiliations:** 1Department of Electronic and Optical Engineering, Space Engineering University, Beijing 101416, China; chenyudi9438@163.com (Y.C.); hongbin_ma@163.com (H.M.); lzwaie@163.com (Z.L.); 2School of Space Information, Space Engineering University, Beijing 101416, China; 1800022837@pku.edu.cn

**Keywords:** group target tracking, unmanned aerial vehicles, maneuver modeling, measurements partitioning, shape estimation, groups’ spawning and combination

## Abstract

Group target tracking (GTT) is a promising approach for countering unmanned aerial vehicles (UAVs). However, the complex distribution and high mobility of UAV swarms may limit GTTs performance. To enhance GTT performance for UAV swarms, this paper proposes potential solutions. An automatic measurement partitioning method based on ordering points to identify the clustering structure (OPTICS) is suggested to handle non-uniform measurements with arbitrary contour distribution. Maneuver modeling of UAV swarms using deep learning methods is proposed to improve centroid tracking precision. Furthermore, the group’s three-dimensional (3D) shape can be estimated more accurately by applying key point extraction and preset geometric models. Finally, optimized criteria are proposed to improve the spawning or combination of tracking groups. In the future, the proposed solutions will undergo rigorous derivations and be evaluated under harsh simulation conditions to assess their effectiveness.

## 1. Introduction

Unmanned aerial vehicle (UAV) swarms have garnered much attention in the military realm, as they offer a means of autonomously and collaboratively fulfilling combat missions using multiple drones [1,2,3,4]. As the demand for effective counter-UAVs is on the rise, the tracking of UAVs is deemed a critical element of the countermeasure [5]. Presently, tracking of UAVs is carried out using multi-target tracking (MTT) with radar or optical detection systems. Several typical systems such as Black Sage Technologies’ Counter Unmanned Aircraft Systems (CUAS) [6], the UK’s Anti-UAV Defense System (AUDS) [7], Dedrone’s Drone Tracker [8], and the GIRAFFE 1X [9] are currently in operation.

However, the increasing number of UAVs in future scenarios presents a more complex tracking challenge. This complexity arises from several factors: (1) When a significant number of highly maneuverable and indistinguishable UAV targets are present, the measurements generated by different targets become indistinguishable, leading to difficulties in forming stable state estimates and trajectories for each target. (2) MTT-based data association is mainly reliant on comparing the similarities between predicted states and the measurement of each target, and is strongly influenced by factors such as the accuracy of predicted states, measurement errors, and clutters; numerous spatially close and maneuverable UAVs lead to poorer tracking accuracy, denser clutter, and the explosion of combinations in association algorithms, rendering data association much less valid and practical. (3) When a UAV swarm moves with a specific pattern, the focus shifts to the overall movement and shape of the swarm, which MTT can only provide individual trajectory information for each target. Treating multiple UAV targets that are spatially close and have similar movement trends as a group and utilizing the trajectory information and spatial distribution characteristics of the group as a guide for counter-UAVs can significantly enhance its efficiency in specific scenarios.

In this regard, group target tracking (GTT) appears to be a perfect fit for UAV swarm tracking. GTT entails estimating and tracking the overall motion state and shape of multiple targets as a group, with its primary steps involving measurement partitioning, shape estimation, state estimation, data association for multiple groups, and detecting and tracking groups’ spawning and combination. Given that the motion and shape characteristics of UAV swarms differ significantly from the target characteristics assumed in previous research scenarios, several aspects of GTT need to be improved to cater to the specific demands of UAV swarm tracking. To avoid redundancy in this paper, related works will be presented with the characteristics of UAV swarms later in each critical technique.

The structure of this paper is as follows: Section 2 presents the basic definition of GTT, its main steps, the inputs and outputs of each step, the evaluation metrics, and compares GTT with MTT to further illustrate the advantages and necessity of GTT for UAV swarm tracking; Section 3, Section 4, Section 5 and Section 6 analyze the limitations of existing GTT works in tracking UAV swarms and propose possible solutions; Section 7 concludes this study.

## 2. Preliminaries

The purpose of this section is to explain the definition and workflow of GTT. This section is divided into three parts. First, the definition of group target tracking is introduced. Then, the workflow of GTT, the input and the output of each step, and the evaluation index are given. Finally, the differences between MTT and GTT are summarized.

### 2.1. Basic Definition

Group target tracking (GTT) is the estimating and tracking of the overall motion and shape state of a group of targets moving in a coordinated manner [10]. GTT can be divided into small and large group tracking based on the number of targets involved. Small group tracking refers to tracking multiple groups with only a few components per group, while large group tracking involves tracking a group with numerous members whose individual members cannot be easily distinguished.

However, it should be noted that the distinction between small and large group tracking based on the number of targets can be controversial, as different target scenarios and detection capabilities may make it difficult to establish a universal rule for distinguishing the two types of tracking. The primary difference between small and large group tracking lies in the way they estimate the group state. Small group tracking involves reprocessing the results of multiple target tracking (MTT), while large group tracking directly estimates the collection of multiple measurements from a holistic perspective without estimating the state of each target. As a result, small group tracking is still based on MTT.

In the context of UAV swarm tracking, the dense distribution and strong mobility of the targets make it challenging for detection equipment to distinguish each UAV target, and data association algorithms of MTT may not work effectively. Therefore, large group tracking is more suitable for UAV swarm tracking. The states of a group typically include the group composition, the motion state of the group centroid (position, velocity, acceleration), and the extended shape of the group.

### 2.2. Workflow and Key Techniques of GTT for UAV Swarm

The workflow of UAV swarms’ GTT mainly includes five key steps, as shown in Figure 1.

Step ①: Measurement Partitioning

Function: The measurements received at the time k are divided into several sets based on a specific criterion, and each set represents a group.

Input: $n_{k}$ measurements at the time k ($n_{k}$ denotes the number of measurements).

Output: $m_{k}$ unlabeled sets at a time k ($n_{k}$ denotes the number of measurements).

Evaluation metrics: Measurement partitioning of UAV swarms is a process of dividing measurements into several measurement sets, which is similar to clustering in machine learning, and evaluating by validity indexes. The validity indexes include the internal index and the external index. The internal index is mainly based on the geometric structure of the dataset in terms of compactness, separation, connectivity, and overlap. Typical internal indexes include the Dunn Index (DI), Davies-Bouldin Index (DBI), and Silhouette Coefficient. When the external information of the dataset is available, the external index evaluates the performance of clustering or partitioning by comparing the matching degree of partitioning with external criteria. Typical external indexes include Normalized Mutual Information (NMI), Rand Index (RI), Jaccard Coefficient (JC), and Fowlkes and Mallows Index (FMI) [11].

Step ②: Groups’ Data Association

Function: The $m_{k}$ unlabeled sets representing different groups at the time k are associated with the corresponding groups’ tracks.

Input: $m_{k}$ unlabeled sets at a time k.

Output: $m_{k}$ labeled sets at a time k.

Evaluation metrics: Generally, the normalized correct associations (NCA) and the incorrect-to-correct association ratio (ICAR) are used to measure the performance of the association algorithm [12]. NCA is the number of correct associations divided by the true number of associations, while ICAR measures the ratio of incorrect to correct associations.
(1)$$\mathrm{NCA}=\frac{\left|\mathrm{CA}\left(\omega \right)\right|}{\left|\mathrm{SA}\left(\omega^{*}\right)\right|}$$
(2)$$\mathrm{ICAR}=\frac{\left|\mathrm{SA}\left(\omega \right)\right|-\left|\mathrm{CA}\left(\omega \right)\right|}{\left|\mathrm{CA}\left(\omega \right)\right|}$$
where SA(ω) represents the set of all associations of the feasibility event and CA(ω) represents the set of correct associations.
(3)$$\mathrm{SA}(\omega )=\left\{\left(\tau ,k_{i}^{\tau },k_{i+1}^{\tau }\right):i=1,\ldots ,|\tau |-1,\tau \in \omega \right\}$$
(4)$$\mathrm{CA}(\omega )=\{\left(\tau ,k,s)\in \mathrm{SA}(\omega ):\tau (k)=\tau^{*}(k),\tau (s)=\tau^{*}(s\right),for\;some\;\tau^{*}\in \omega^{*}\}$$
where $k_{i}^{\tau }$ represents the time of the $i^{th}$ group on track τ. τ(k) represents the association between the centroid of the group and the track τ at a time k. ω denotes the set of associated tracks. $\omega^{*}$ denotes the set of true tracks. |A| is the cardinality of the set A.

Step ③: Estimation of a Group’s Motion State

Function: The position, velocity, and acceleration of a group’s centroid are used to describe the motion state of the group. Using precise motion models and filtering algorithms, the motion state of each group is estimated from the centroid of the groups’ measurements.

Input: measurements $\left\{z_{i,1}^{k},z_{i,2}^{k}\ldots z_{i,I}^{k}\right\}$ of the group at a time k.

Output: $i^{th}$ group’s motion state estimation ${\hat{x}}_{i}^{k}$ at time *k* and state prediction ${\hat{x}}_{i}^{k|k+1}$ at time *k* + 1.

Evaluation metrics: There are two evaluation metrics, one considering single group tracking and the other considering multiple group tracking. The root means square error (RMS) of position and velocity estimation are used to investigate the effectiveness of motion models and filtering algorithms in single group tracking.
(5)$$\left\{ \begin{array}{l} RMSE\left(x\right)=\sqrt{\frac{1}{M}\sum_{i=1}^{M}{\left(x_{j}-{\hat{x}}_{j}\right)}^{T}\left(x_{j}-{\hat{x}}_{j}\right)}\\ RMSE\left(v\right)=\sqrt{\frac{1}{M}\sum_{i=1}^{M}{\left(v_{j}-{\hat{v}}_{j}\right)}^{T}\left(v_{j}-{\hat{v}}_{j}\right)} \end{array}\right.$$
where M represents the number of Monte Carlo simulations, and j is the index of the number of simulations. Optimal sub-pattern assignment (OSPA) defines a measure distance in the state space, which is used to evaluate the error size of the real and estimated tracks of multiple groups [13]. By adjusting the distance sensitivity parameter p and association sensitivity parameter c, the tracking performance under different attention angles was investigated.
(6)$${\bar{d}}_{p}^{\left(c\right)}={\left(\frac{1}{{\hat{m}}_{k}}\left(\underset{\pi \in \prod {\hat{m}}_{k}}{\min}\sum_{i=1}^{m_{k}}d^{\left(c\right)}{\left({\hat{x}}_{i}^{k},x_{\pi \left(i\right)}^{k}\right)}^{p}+c^{p}\left({\hat{m}}_{k}-m_{k}\right)\right)\right)}^{1/p}$$
where the actual state of each group is represented by $x_{i}^{k}$, ${\hat{x}}_{i}^{k}$ is the estimated state of each group, i is the serial number of the group, and k is time. The real number of groups is denoted by an estimated number ${\hat{m}}_{k}$. $\prod {\hat{m}}_{k}$ represents the permutations on selecting $m_{k}$ states from the set $\left\{{\hat{x}}_{1}^{k},{\hat{x}}_{2}^{k},\ldots ,{\hat{x}}_{{\hat{m}}_{k}}^{k}\right\}$.

Step ④: Estimation of the Group’s Shape

Function: The shape of the group is described by a specific model whose parameters are estimated from the measurements of the group. Moreover, filtering algorithms are used to smooth shape estimation.

Input: measurements $\left\{z_{i,1}^{k},z_{i,2}^{k}\ldots z_{i,I}^{k}\right\}$ of the group at time k.

Output: $i^{th}$ group’s shape estimation at time *k* (corresponding models and parameters).

Evaluation metrics: The concept of area error is introduced to evaluate the degree of mismatch between the estimated value and the real shape [14]. The difference in symmetry between two shapes is expressed as the union of shapes minus their intersection, which is denoted by Δ:(7)$$\hat{S}\Delta S_{G}=\left(\hat{S}\cup S_{G}\right)\backslash \left(\hat{S}\cap S_{G}\right)$$
where $\hat{S}$ represents the group’s shape estimation and $S_{G}$ represents the group’s real shape. The normalized area error ε is expressed as:(8)$$\varepsilon =\frac{\left\|\hat{S}\Delta S_{G}\right\|}{\left\|S_{G}\right\|}$$
where ‖ ⋅ ‖ denotes the area of the area. When the estimated shape is 3D, volume error is introduced instead of area errors.

Step ⑤: Tracking of Group’s Spawning and Combination

Function: According to the motion state and shape overlap of groups, the spawning and combination of groups are detected and recorded based on a specific criterion, whose results act on groups’ data association at a time k+1. The track initiation of new groups is realized by a specific state transition.

Input: all groups’ shape and motion state estimation at time *k*.

Output: modified groups’ states’ new labels, states, and eigenvalues for association at time *k* + 1 for the groups generated by spawning and combination; termination of groups’ tracks involved in spawning and combination.

Evaluation metrics: Since the result of tracking of group’s spawning and combination includes the modification of the groups’ states and the association acting on the group, it is mainly evaluated comprehensively from the number of groups estimated and the tracking error of each group instead of a specific metric [15].

### 2.3. Contrast between MTT and GTT

As an extension of MTT, GTT has distinct differences from MTT in terms of workflow, output results, and applicable scenarios. Contrast among single-target tracking (STT), MTT, and GTT is shown in Table 1.

## 3. Measurement Partitioning for UAV Swarms

### 3.1. Related Works

Measurement partitioning methods can be categorized into two types: distance partitioning (including derived distance partitioning) and cluster partitioning. In distance partitioning, two measurements belong to the same group when the distance between the two measurements is less than a specific threshold. By traversing and evaluating all measurements, the measurements belonging to the same group are selected. Classical distance partitioning includes Granström’s distance partitioning and sub-partitioning [16], distance partitioning based on SNN similarity [17], Geng’s clustering matrix method [18], Wang’s distance partitioning based on coordinates transformations and distance differentiations [19], and generalized distance partitioning [20].

In cluster partitioning, measurements are considered to form a group if a combination of these measurements is more akin to a group. The criterion is based on whether the high-order features of the measurements meet the specific threshold. At present, the applications of clustering partitioning include K-means clustering [16], EM clustering [21], spectral clustering [22], DBSCAN clustering based on SNN similarity [17], adaptive resonance theory partitioning [23], and shape selection partitioning [24].

### 3.2. Challenges

① Measurements of Non-Convex Distribution

When performing specific tasks, UAV swarms presented complex and irregular shapes, such as unique formation shapes or highly clustered shapes, as shown in Figure 2 (screenshot of Perdix telemetry video). Prototype clustering’s such as K-means clustering, EM clustering, and AP clustering, perform poorly for measurements of non-convex distribution, as shown in Figure 3.

② Unpredictable Number of Groups

Prototype clustering methods such as K-means clustering, EM clustering, AP clustering, and spectral clustering, require the number of sets formed by partitioning to be set in advance, making them impractical for target tracking. This is because the number of groups formed by non-cooperative targets is unknown, and the accurate number of clusters cannot be manually set. Furthermore, the number of clusters in the field of view changes as new targets enter the field of view, or clusters combine or spawn. To identify the number of clusters, methods such as the elbow method, contour coefficient method, and Calinski Harabasz value method are commonly used [25]. However, determining a reliable number of clusters for the same batch of measurement points or data used for analysis requires several Monte Carlo simulations to obtain an average value, which is computationally expensive and time-consuming. Thus, the unpredictable number of clusters poses a significant challenge for prototype partitioning in real-time tracking systems.

③ Unevenly Distributed Groups

During the task execution of UAV swarms, the distribution density of UAVs in different swarms may vary due to the different types of UAVs or the different maneuvering tasks. Therefore, it may be possible for different groups in the field of view to produce measurements that are distributed with different densities. Density clustering and distance partitioning are sensitive to this issue. In the distance partition, only a threshold is used to determine whether two measurements belong to the same group. When several groups with different densities are close to each other, improper selection of distance threshold will cause unreliable clustering [17] and interfere with subsequent tracking.

### 3.3. Solutions: Partitioning Based on OPTICS Clustering

OPTICS algorithm is proposed to address the limitation that DBSCAN cannot cope with different densities. OPTICS is insensitive to the distribution density and can be used to partition measurements of non-convexly distributed densities when the number of clusters is unknown. The algorithm involves two parameters: MinPts, where ε denotes the distance threshold that constitutes the neighborhood, and MinPts denotes the minimum number of measurements that constitute the neighborhood in the partition.

OPTICS, however, returns an ordered list of objects. It is necessary to select the appropriate ε according to the ordered list, then obtain the partition result by DBSCAN clustering. The selection of ε in previous applications was based on manual intervention. However, for group target tracking, measurement partitioning of each frame by manual intervention is unpractical. Therefore, a method based on peak identification is proposed to automatically select an appropriate ε. After smoothing the data in the ordered list, significant peaks are screened out and the ε is set just short of the minimum peak. Algorithm 1 shows the pseudo-code for the automatic partitioning. This solution mainly refers to the OPTICS clustering and DBSACN clustering. Due to the practical needs of the application, an ε selection method based on peak identification is introduced to realize automatic partitioning. Peak identification in pseudo-code is temporarily represented by the ‘findpeaks’ function and needs further research on the characteristics of measurement distribution of UAV swarms.
**Algorithm 1** automatic measurement partitioning. Automatic partitioning based on OPTICS clustering.**Input:** measurement set $X\left\{z_{1}\left(coord\right),z_{2}\left(coord\right),\ldots ,z_{N}\left(coord\right)\right\},eps,MinPts$**Output:** measurements with group labels $X^{\prime }\left\{z_{1}\left(coord,m_{1}\right),z_{2}\left(coord,m_{2}\right),\ldots ,z_{N}\left(coord,m_{N}\right)\right\}$1:ordered_list = OPTICS(X, eps, MinPts)2:ordered_list’ = smooth_filter(ordered_list)3:peaks = findpeaks(ordered_list’)4:eps_new = min(peaks)5:clusterID = −16:*k* = 17:**for** i = 1, 2, … to N **do**8:  j = get_z_id(ordered_list(i))9:  **if** reach_dist(j) > eps_new or reach_dist(j)= =UNDEFINED **then**10:   **if** core_dist(j) < eps_new or core_dist(j) ≠ UNDEFINED **then**11:    clusterID = *k*12:    *k* = *k* + 113:    $m_{j}$ = clusterID14:   **else**15:    $m_{j}$ = −116:   **end if**17:  **else**18:   $m_{j}$ = clusterID19:  **end if**20:**end for**

## 4. Groups’ Data Association and Motion States Estimation for UAV Swarms

### 4.1. Related Works

There are two types of methods for group data association. The first type extends the data association methods used in multi-target tracking. Improved JPDA [26,27] and PMHT [28] algorithms have been proposed to fit GTT. The second type of method is association in Random Finite Sets (RFS), which includes particle labeling [29], association of estimation and track [30], association based on fuzzy clustering [31], and association based on cross entropy [32].

Tracking a group’s centroid is similar to tracking a single target. The process involves motion modeling of estimators and adaptive filtering. There are two ways to model motion. The first is to use a hypothetical motion model to fit the moving process of the target when there is no prior information. These models assume that targets’ maneuvering is caused by non-zero acceleration [33]. Typical models include constant velocity, constant acceleration, polynomial model, Singer model [34], current statistical model [35], and Jerk model [36]. To cope with complex maneuvering of targets, researchers have proposed the multiple-model algorithm [37], which estimates the motion state of the target by using multiple models (MM) in parallel and obtaining the state of the target by weighting the results of different models. To enhance models’ robustness and reduce computational complexity, researchers have also proposed the interacting multiple-model (IMM) algorithm [38], the IMM with variable structures [39], and the adaptive IMM [40]. The second way is to construct a deterministic model to describe the change of target acceleration and relate the acceleration change to the observed quantity by utilizing prior dynamic information of the target. The typical model is the gravitational model of orbital targets in space.

The filtering algorithms for target tracking can be divided into linear filtering and nonlinear filtering. For the state filtering of a UAV swarm with a nonlinear motion equation and observation equation, nonlinear filtering is a better choice. Nonlinear filtering includes classical extended Kalman filtering [41,42], untraced Kalman filtering [43], particle filtering [44], and random finite set filtering [45]. Currently, the research of GTT state filtering mainly focuses on RFS theory. The RFS filter assumes two hidden layers of the target state: the first is the state of each target, and the second is the state of each group. The second hidden layer is a random set of some individual target states in the first hidden layer [45]. Bayesian filtering based on RFS avoids the combination problem caused by direct data association in the tracking process and makes the track association and filtering decouple. Due to the unsolved set integration in Bayesian filtering based on the RFS framework, the suboptimal solution is used to approximate the standard Bayesian filtering in practice. Currently, RFS filters used in GTT mainly include probability hypothesis density (PHD) filters [16,21], Cardinality PHD (CPHD) filter [46], Multi-Bernoulli (MB) filter [47], Labeled multi-Bernoulli filter (LMB) [48,49], and Generalized LMB (GLMB) filter [47]. However, the output (including prediction) of RFS filtering is a discrete, unordered, set-based state estimate. This prediction is not what is traditionally understood as predicting what state a particular target will be in at the next moment. If the results of RFS filtering are to be used to form trajectories for multiple groups or to drive radar beam scanning, radar signal processing, and optical system angle tracking, data association needs to be performed on the RFS results.

### 4.2. Challenges

There are two main challenges for group association and motion state estimation:

① High Maneuverability of UAV Swarms

UAVs can maneuver with large loads, including direct flight with significant acceleration, high-speed turns, fast S-shaped maneuvers, and maneuvers with variable acceleration for specific mission requirements [50]. Moreover, rotary UAVs and fixed-wing UAVs’ maneuvering characteristics are also very different. This greatly impacts the motion state estimation and data association of UAV swarms.

Theoretically, MM algorithms can approximate various motions, which have to accumulate sufficient observation data to form proper estimations of the movement models, mainly when there are heavy observation noise and nonlinear approximation errors [51]. In traditional MM algorithms, the correct model estimated by previous observations is always behind the current target state, which results in a significant decline in tracking performance when the maneuver happens [51,52,53]. When the UAV swarm maneuvers frequently and it is difficult to accumulate enough observation between each maneuver to accurately estimate the current motion model, the tracking error caused by the model-estimation delay of MM algorithms will become intolerable. Moreover, the movement of UAV swarms has complex and variable acceleration rather than a combination of several fixed motion models. If the model combination in MM algorithm cannot adequately describe the maneuver of the UAV swarm, it will also cause model error and tracking error.

Data association is based on the relationship between state predictions and current measurements. When the strong mobility of the UAV swarm causes tracking error, the association of close and maneuvering groups tends to fail, easily leading to the loss of targets.

② Clutter and Noise Interference

In fact, after measurement partitioning, there may still be cluttered or false measurements in the field of view. In the existing association algorithms, the only available association feature is the spatial position. Without multi-dimensional characteristics of groups, the association will likely be interfered with by clutters.

### 4.3. Proposed Solutions

#### 4.3.1. Deep-Learning-Aided Multiple UAV-Dynamic-Model Tracking

Traditional multi-model tracking models the motion pattern of a target as a combination of multiple hypothetical motion states, which is suitable for tracking targets that have no prior information and are less maneuverable. However, UAV swarms maneuver frequently in some situations, and existing multi-model approaches generate model estimation delays when maneuvers occur, generating tracking errors [53]. The tracking performance degrades significantly when the maneuvers are too frequent and do not match the initial maneuver parameter settings.

The existing UAV dynamics model is constructed from the perspective of controlling the UAV, and the factors affecting the UAV maneuver are thrust and axial deflection. The maneuver model is derived from the control model of the UAV itself for the third view observation. For the non-cooperative UAV targets, there are still time-varying unknown parameters in the model, even if the back-and-forth transfer relationship of the motion state is constructed. With similar problems, the Singer model considers the target acceleration as a correlated noise, which assumes that the target acceleration a(t) is a zero-mean stochastic process with exponential autocorrelation, and its time correlation function is in the form of exponential decay. Based on the above idea, the covariates to be observed and estimated in the UAV motion model are similarly modeled as exponentially autocorrelated zero-mean stochastic processes.

Taking a quad-rotor UAV as an example, the centroid’s mechanical model is shown in Figure 4. By modeling the normal acceleration, pitch angle, and azimuth angle of the fuselage as Markov processes, the UAV state tracking is transformed into the trend of estimating the above variables through observation. Moreover, according to the characteristics of the UAV, reasonable maneuvering frequency and variation range can be set for these variables.

In Figure 4, x, y, and z represent the position of the group’s centroid; v represents the speed value of the centroid relative to the ground; γ is the track deflection angle; ψ is the track pitch angle; $n_{x}$ is the tangential load in the direction of the track; $n_{z}$ is the normal load perpendicular to the direction of motion; μ is the roll angle that controls $n_{z}$.

Since the type of the UAV swarm to be tracked cannot be known in advance, the corresponding motion model needs to be determined by motion feature extraction for tracking. The deep learning method used in [51,52] extracts the motion features of the tracked target. The results of the bi-directional LSTM network are then used to select the compatible motion model for state tracking. Therefore, this paper suggests using a deep-learning-based method for UAV motion feature discrimination. However, our proposed motion model differs significantly from the one in [52], and the envisioned maneuvering states of the UAV swarm also differ greatly, requiring different feature extraction. Therefore, we used an LSTM network to assist the motion state filtering of the UAV swarm based on the existing framework, with significant differences in the embedding and connection layers compared to [52]. The architecture of the method is shown in Figure 5, and the training process of the network is shown in Figure 6. The different motion models correspond to different UAV maneuvering and can be obtained by extensive prior modeling.

#### 4.3.2. Groups’ Data Association Based on Multi-Feature Fusion

A group has multiple characteristics, and the reliability of data association can be improved by considering the degree of similarity of multiple features. The centroid position, the number of measurements, the group’s average Doppler frequency, and the group’s shape are used as the reference features, and the corresponding feature vectors are constructed. Minkowski distance, cosine correlation coefficient, Pearson correlation coefficient, or Mahalanobis distance, can be used to calculate the similarity between current and predicted group characteristics. An algorithm similar to the nearest neighbor association can be used to associate groups with the greatest similarity. Furthermore, to improve the anti-jamming ability of association, PDA, JPDA, PMHT, and particle labeling, can be modified using the above multi-feature similarity.

## 5. Groups’ Shape Estimation for UAV Swarms

### 5.1. Related Works

According to the complexity and application scope of the shape estimation model, it can be divided into simple geometric shape estimation (such as circle, ellipse, rectangle, etc.) and complex shape estimation. According to the random matrix model proposed by Koch, the measurement is an approximately elliptic distribution generated around the center of a group according to some random distribution [54]. In order to improve the accuracy and robustness of the model, Feldmann and Lan et al. deduced the extended state evolution model which can reflect the shape, size, and direction of the group with time [51,54]. The random hypersurface model (RHM) performed Cholesky decomposition on the matrix, describing the expanded shape of the ellipse [55,56]. The elements of the decomposed lower (upper) triangular matrix were taken as state elements and combined with the target motion state to form an augmented vector describing the group target state. The joint estimation of the motion state and the extended state of the group target centroid is realized by the estimation of the augmented vector. In addition, based on the idea of an augmented vector, a star-convex model was proposed [57]. The rectangular parameter method [58] and interval box method [46] are used to model the shape of the group target as a rectangle. In complex shape estimation, the object studied was mainly the plane shape corresponding to 2D measurements, including the Level-set RMH [14], the multi-elliptic model [59,60], and the extension-deformation approach [61]. Multi-ellipse models are divided into overlapping multiple ellipses at the same centroid and splicing multiple ellipses at different centers. Multi-elliptic models with different centers can describe more complex shapes. However, the models’ limitation is that the relative position of each center is fixed, which no longer changes in real-time and can only be used to describe shapes with fixed structures.

### 5.2. Challenges

UAV swarms present complex and diverse formation or aggregation patterns for specific tasks. Figure 7 shows the demonstration of NAVAIRs Perdix in which the swarm presents a linear formation, a contracting aggregation, a closed circular formation, and a non-closed circular formation. To analyze the purpose and intention of UAV swarms, a suitable model is required to describe the swarm’s time-varying shape, especially with three-dimensional radar measurements. Among the existing shape models, only the ellipse model, rectangle model, and multi-ellipse model can be directly extended to three dimensions. In the application of these models, it is assumed that the shape structure of the swarm target does not change significantly, and the measurements are distributed according to the same shape structure within the tracking range. In simpler terms, existing methods can hardly accurately describe the variation between different distributions of group shapes. Furthermore, models that can describe shapes with a certain degree of complexity are limited to two dimensions and are difficult to directly extend to three dimensions, such as Level-set RMH [14] and the extension-deformation approach [61].

### 5.3. Proposed Solutions: 3D Shape Estimation Based on Fitting of Key Points

Estimating human posture based on monocular images is a new research topic in computer vision. The main research work in this field is to extract human features in the image by inputting the human RGB image data recorded by a single camera, then to estimate the 3D model of the output human posture combined with a specific human posture model [62]. In order to extract the corresponding features to represent the human posture, the most critical content is how to use the model to represent the complex human posture.

Inspired by the estimation of human posture, this paper proposes a method for estimating the three-dimensional shape of group targets based on key points fitting [63,64,65,66]. The shape estimation includes two basic steps. Firstly, key points are extracted based on the characteristics through a deep network. Then, different key points are matched with preset 3D geometric shapes. The shape estimation is formed by parameters of shape and attitude, as shown in Figure 8. Compared with the previous models, the shape estimation method based on the key structure points can reflect the inner hollow state of the group. It can also reflect the formation state of the group to a certain extent through the key structure points. However, it is also necessary to preset the 3D shape in advance to cover the possible distribution.

## 6. Tracking Spawning and Combination of UAV Swarms

### 6.1. Related Works

Granström was the first to propose a complete framework for tracking group spawning and combination [15]. This algorithm only considers simple spawning and combination scenarios and adopts the target assumption based on the random matrix model. Based on the above framework, Gan uses a δ-generalized labeled multi-Bernoulli filter to rederive the spawning criteria and state transfer under three classical spawning modes, which proves to be more accurate and efficient [67,68]. Geng proposed a second association separation detection model of pre-set wavelet gate within the association gate [18]. The basic idea is to generate a wavelet gate for each measurement within each group, which is used to predict the possible range at the next moment. When the location of the in-group measurements is not within the range predicted by the wavelet gate, the model considers these measurements to be false measurements or those with a tendency to separate. After that, the algorithm adjusts the weights of the anomaly measurements to reduce the impact on the current group state.

### 6.2. Challenges

① More Than Two Groups Involved in Spawning and Combination

In practice, the spawning of a group or the combination of multiple groups may involve more than two groups. The existing models only consider the state of one group spawning into two groups and two groups merging into one group. A universal model is needed to describe it.

② Spawning/Combination Criterion with Groups’ Complex 3D Shapes

Granström’s algorithm believes that the measurements of the group target follow the GIW distribution, which means the shape of the group is oval. Based on this, the algorithm designs a criterion for combination detection. In this criterion, when the elliptic shape of two groups overlaps and the difference of the group velocity vector is less than a specific threshold, the combination of the groups is considered to occur. When a group is modeled with a complex 3D shape, a more accurate algorithm is needed to the shape overlap.

### 6.3. Proposed Solutions: New Criterion for Tracking Groups’ Spawning and Combination Modeled with Complex Shapes

This paper proposes a multi-group combination tracking model based on measurements reclustering in the cross-tracking gate. This method is a further development of the model proposed by Granström. The main improvement is to repartition the measurements in the tracking gate where the group targets with non-elliptic distribution are merged. When the number of groups generated by clustering is less than the original number of tracking gates, it is considered that group merging behavior begins to occur. At the same time, the model improves the description of detection and state passing when multiple groups merge simultaneously. The tracking process of multiple groups’ combination is shown in Figure 9.

In Granström’s model, it is unclear what physical characteristics determine the group spawning among his model, nor can it be extended to the application scenario of spawning into multiple groups. This paper proposes a group spawning tracking model based on tracking gate range variation characteristics. An adaptive tracking gate is set to automatically adjust the size range by following the distribution and motion trend of the measurements within the group. The range-adjusted change rate is taken as the characteristic of target spawning in the dense group. When the change rate is greater than a threshold, it is considered that the target in the group may have a spawning trend. Then, all measurements involved in the tracking gate are repartitioned. When the number of groups repartitioned is larger than one, terminating the track of the former group will occur. Then, the state estimation and follow-up tracking of the newly generated groups are carried out. This model applies to a group spawning into multiple groups, as shown in Figure 10.

## 7. Conclusions

Group target tracking (GTT) is an efficient scheme for tracking various targets, including orbital debris swarms, extended targets in autopilot, and biomes. However, GTT faces challenges in accurately tracking UAV swarms due to their strong mobility and complex distribution. This paper proposes possible solutions to this issue including OPTICS-based measurement partitioning, multi-feature fusion for anti-interference data association, deep-learning-aided multiple UAV-dynamic-model tracking, key point extraction for 3D shape estimation, and optimized criteria for tracking group spawning or combination. The measurement partitioning mainly refers to the existing research work and only needs application adaptation. The centroid tracking and shape estimation of the UAV swarm refer to the research framework in other fields; they will be first applied in GTT and required a major redesign. Proposed tracking spawning or combination of groups is an improvement on existing methods in GTT.

To validate these solutions’ feasibility and effectiveness, rigorous derivation of formulas and more complex simulations need to be conducted that consider various group distributions and densities, strong mobility, and complex combination/spawning. These proposed solutions have the potential to enhance the effectiveness of counter-UAVs and improve tracking accuracy and precision. Further research and testing are necessary to validate their effectiveness in practical applications.

## Figures and Tables

**Figure 1 sensors-23-04465-f001:**
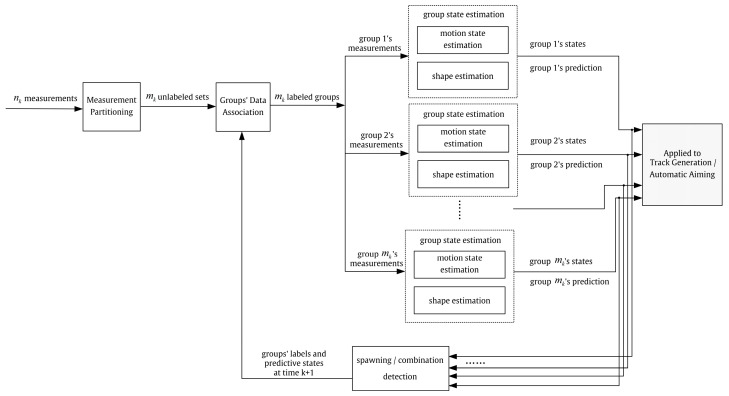
Workflow of UAV swarms’ GTT.

**Figure 2 sensors-23-04465-f002:**
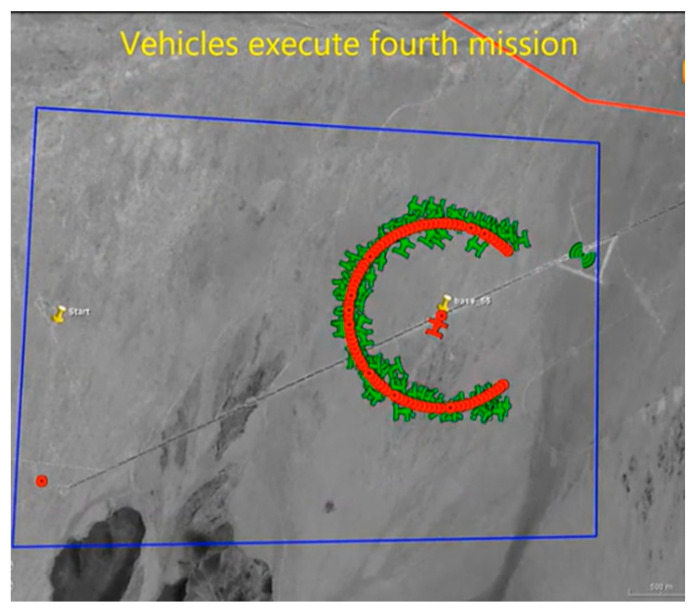
Screenshot of Perdix telemetry video, where the red circles represent the expected locations of UAVs, while the green airplane icons represent the current locations of the UAVs.

**Figure 3 sensors-23-04465-f003:**
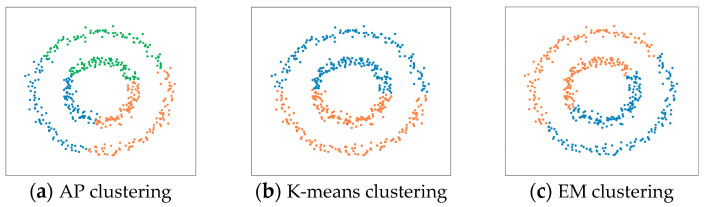
Clustering of measurements with non-convex distributions: (**a**–**c**) are the results of AP clustering, K-means clustering, and EM clustering on the concentric circle distribution, respectively; (**d**–**f**) are the results of AP clustering, K-means clustering, and EM clustering on the approximate crescent moon distribution, respectively.

**Figure 4 sensors-23-04465-f004:**
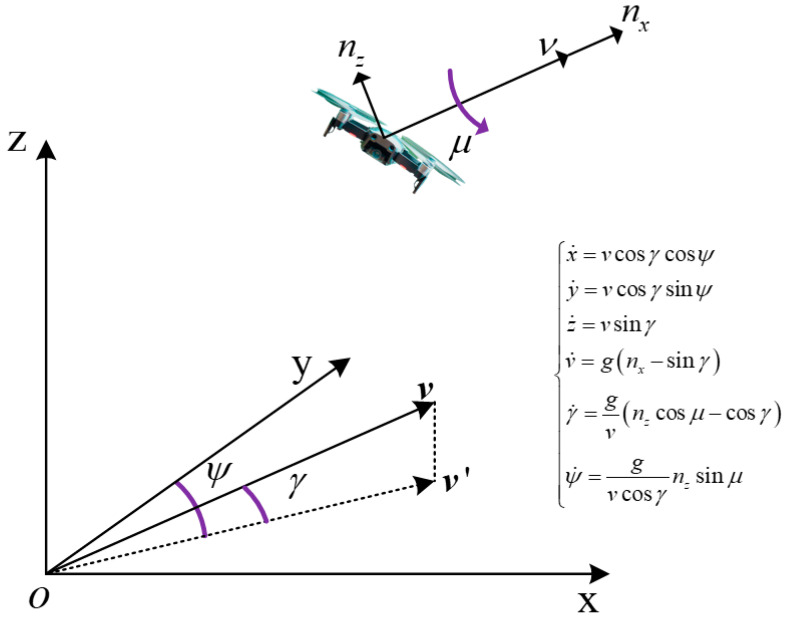
Mechanical model of the centroid of the quad-rotor UAV.

**Figure 5 sensors-23-04465-f005:**
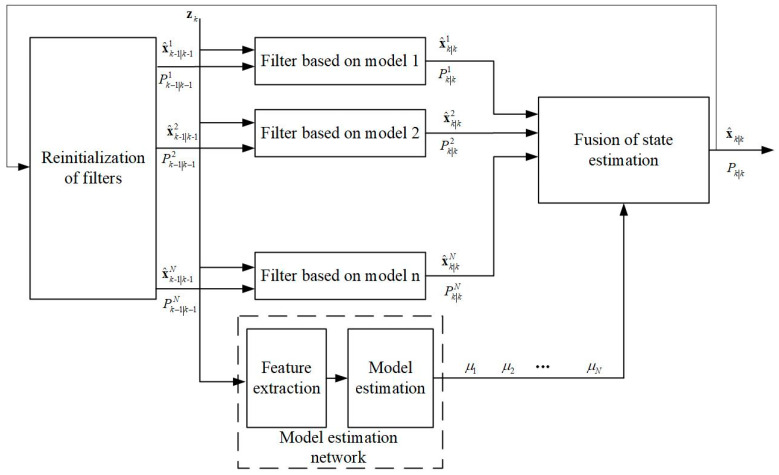
Architecture for centroid tracking based on deep-learning-aided multiple UAV-dynamic-model.

**Figure 6 sensors-23-04465-f006:**
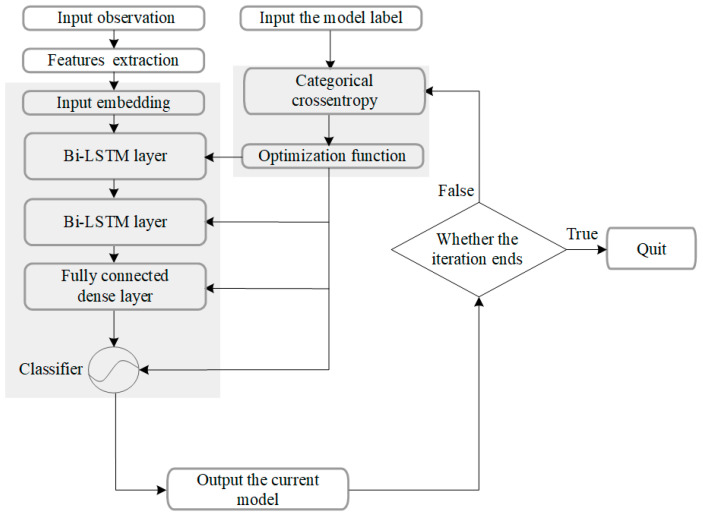
Training process of the model estimation network.

**Figure 7 sensors-23-04465-f007:**
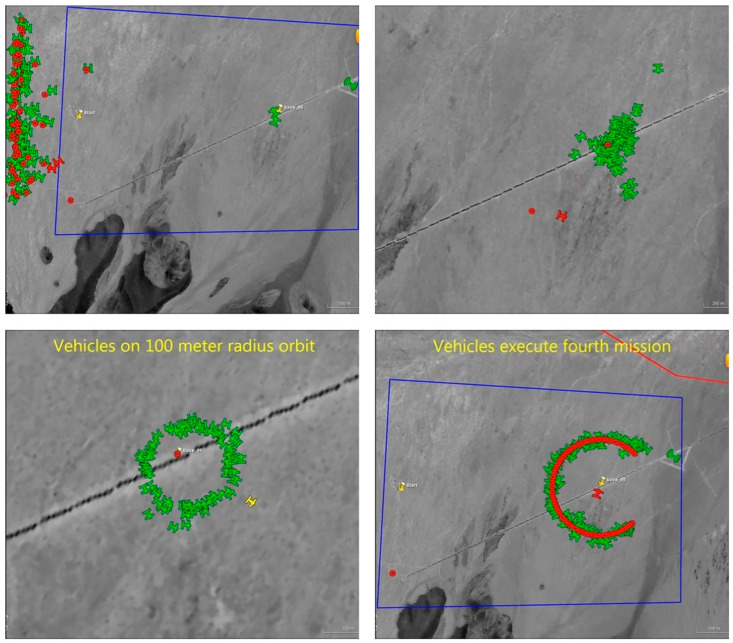
Screenshots of Perdix telemetry video, showing UAV swarms with different distribution shapes: a linear distribution in the top left, a clustered distribution in the top right, a closed ring distribution in the bottom left, and an unclosed ring distribution in the bottom right.

**Figure 8 sensors-23-04465-f008:**
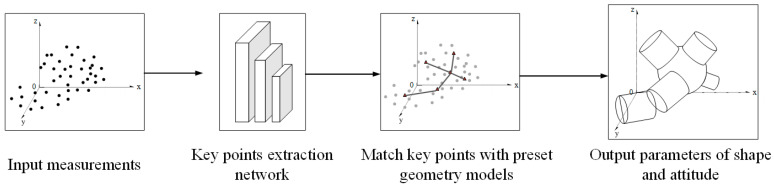
3D shape estimation based on fitting of key points.

**Figure 9 sensors-23-04465-f009:**
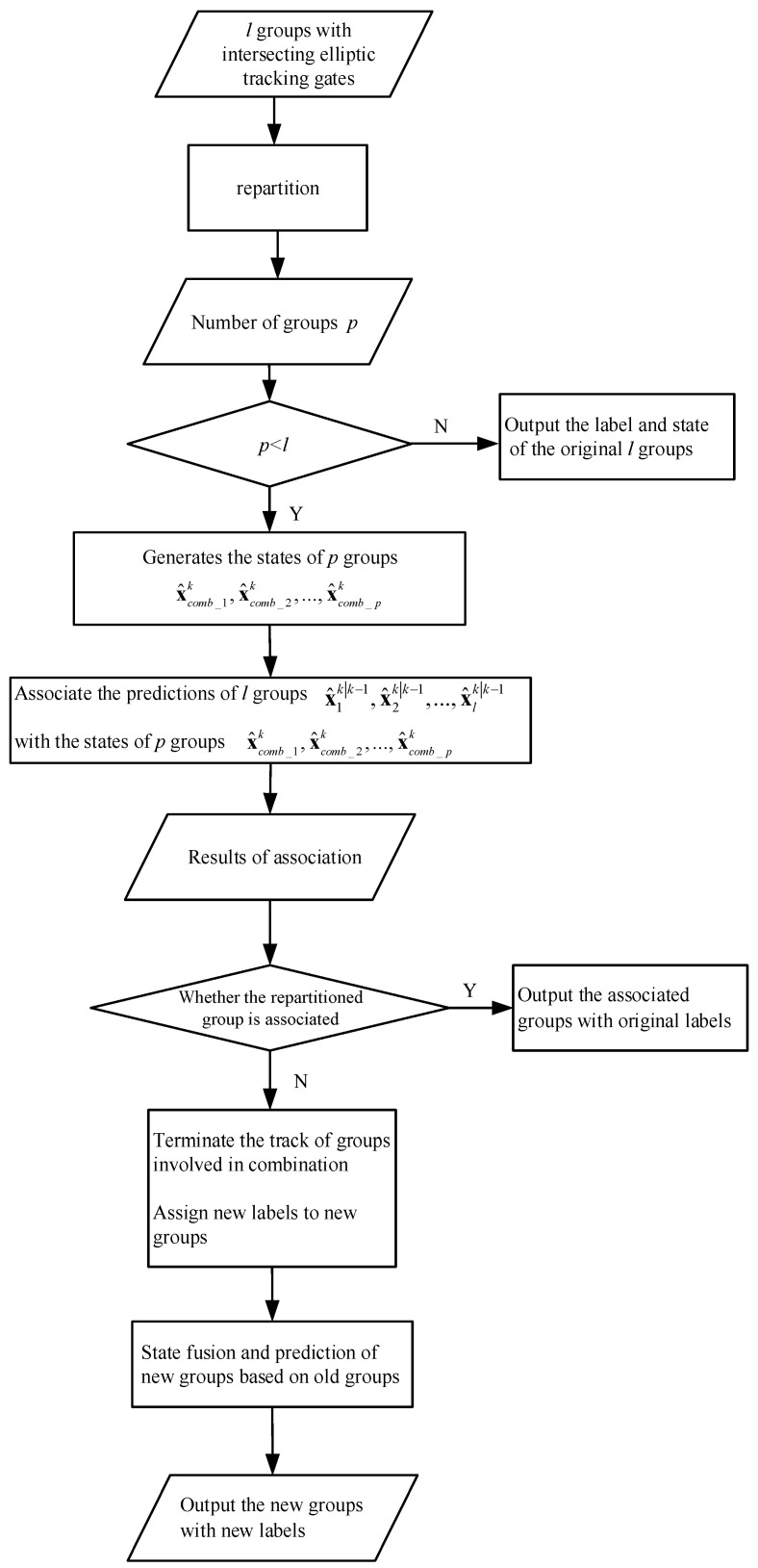
Tracking of multiple groups’ combination.

**Figure 10 sensors-23-04465-f010:**
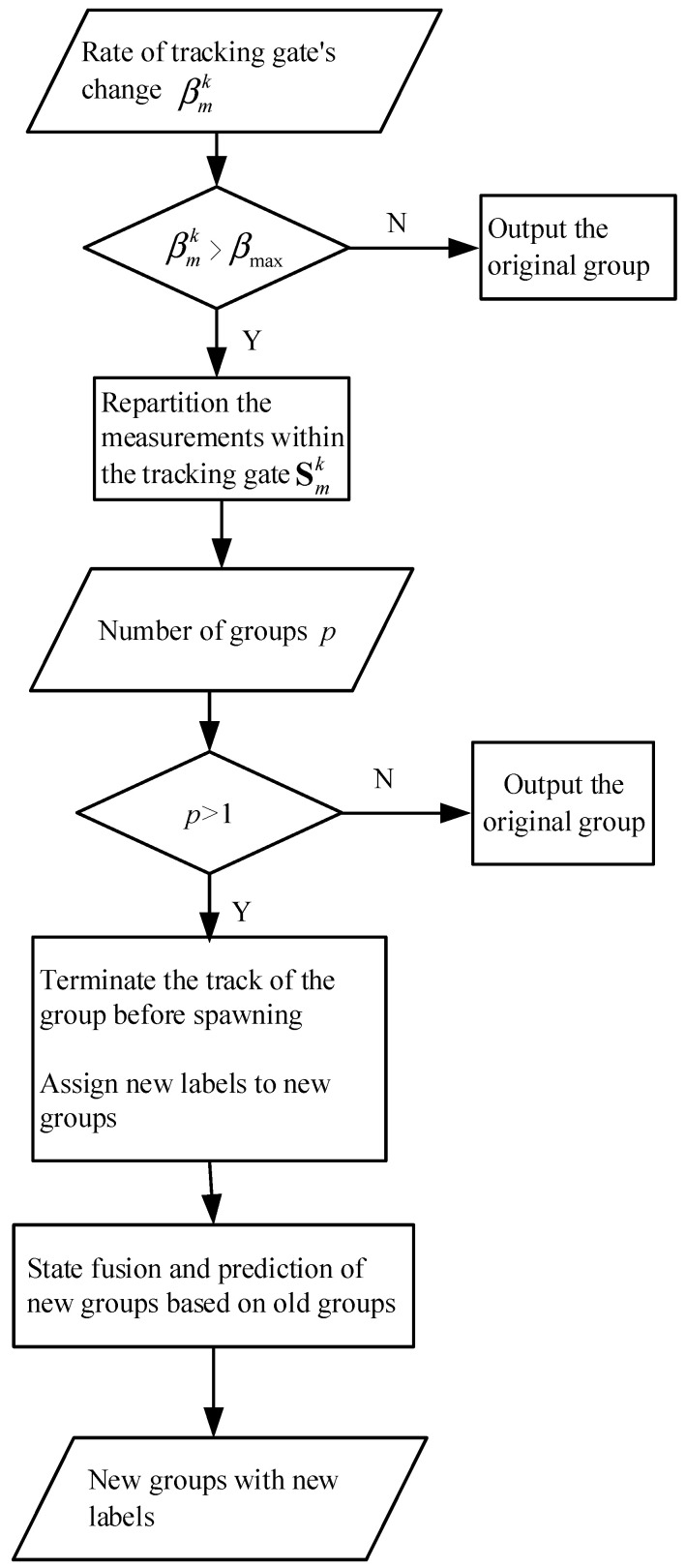
Tracking of multiple groups’ spawning.

**Table 1 sensors-23-04465-t001:** The contrast among STT, MTT, and GTT.

Aspects	STT	MTT	GTT
Number of Targets	1	l	l
Number of Measurements	1	n,n≥l and n≈l	n,n≠l
Number of States	1	l	m,m<n, m≪l
States of output	Centroid’s motion state	Centroid’s motion state of each target	Centroid’s motion state and shape state of each group
Workflow	State filtering	Data association + State filtering	Partition + group association + state filtering
Technical Focus	Target motion modeling and filtering	Data association	Measurement partitioning, groups’ shape estimation, spawning/combination detection
Applicable Scenarios	One target in the field of view	When tracking multiple targets, data association algorithms are effective	Data association of MTT fails Groups’ states are objects of interest

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
