# Peer review of "Group Target Tracking for Highly Maneuverable Unmanned Aerial Vehicles Swarms: A Perspective"

_sensors, 2023, doi:10.3390/s23094465_

Round 1

Reviewer 1 Report

The paper is about group target tracking (GTT) for unmanned aerial vehicles (UAVs) swarms. The paper introduces the definition and workflow of GTT, and compares it with multi-target tracking (MTT). It also proposes potential solutions to improve the performance of GTT for UAV swarms: This paper introduces a density clustering method based on OPTICs algorithm to divide UAV swarm measurements with non-uniform and arbitrary contour distributions. In this paper, a deep learning method is proposed to model the maneuver of UAV swarm to improve the accuracy of centroid tracking. It proposes a method to estimate the three-dimensional (3D) shape of a group by applying key extraction and preset geometric models. The paper proposes the optimized criteria to improve the spawning or combination of tracking groups.

In the following, some comments are given.

1. The author proposes a clustering method based on the OPTICs algorithm, named OPTICS Clustering with Parameter Optimization, but its basic method is DBSCAN. What's more, combining optical algorithms with DBSCAN is not the first time. However, it seems that the authors have misattributed it to their own innovations.

2. In section 4.1, the authors make a point that existing RFS filters lack target state prediction, which is suitable for offline data processing and track generation. This seems a misunderstanding of the RFS filter. First, RFS has the predict-update step, as usually in the common filters. Second, the RFS filter is obviously not "the offline data processing filter". Also, for RFS filters, you can pay attention to the latest developments, such as the multi-sensor RFS fusion approach.

3.In section 5.2, the author says in line 427 that UAV does not obey the assumed probability distribution in practice, which is too absolute. If it does not match reality, please remove these assumptions.

4.Figures should be labeled and explained so that the reader can understand them easily. For example, in Figure 8, the lack of annotations makes it difficult to understand.

5.There are still some irregularities and grammatical errors in the article. Please correct them carefully

Reviewer 2 Report

In the paper, the authors discuss challenges in group target tracking for highly maneuverable unmanned vehicle swarms and propose solutions to these challenges.

The review of UAV group tracking challenges is written relatively well and logically organized. However, the description of the solutions has many deficiencies.

Firstly, it's difficult to understand if a solution is a novel work by authors or if authors present an existing work/method. E.g. in Section 4.3.1 authors mention two LSTM-based methods to extract motion features. Then, the authors present LSTM-based network architecture in Figure 7. It's unclear if the authors developed this architecture and the network training process or if it's based on existing work. It's unclear whether the algorithm in Figure 4 (Optimized measurement partitioning with OPTICS clustering) is developed by authors or is taken from an existing work. Also, the OPTICS method is not properly referenced.

Some methods presented as solutions to identified challenges are described only in general and miss details. E.g. It's not clear what is the input to the network depicted in Fig. 7, what is the role of features extraction and input embedding steps, and what is the size/dimensionality of LSTM and fully connected layers. Also, there're no details on the network training process nor a training and evaluation dataset.

Secondly, there's no evidence or quantitative results proving that the proposed solutions work as intended. If any of these solutions were developed/invented by authors, they should be properly evaluated and compared with the state of the art. If these solutions are taken from the literature, it should be made clear in the paper what their source is.

Round 2

Reviewer 2 Report

The revised version of the paper addresses my concerns raised during the review of the initial version. Therefore I recommend to accept the paper.